# Protein Disulfide Isomerase 4 Is an Essential Regulator of Endothelial Function and Survival

**DOI:** 10.3390/ijms25073913

**Published:** 2024-03-31

**Authors:** Shuhan Bu, Aman Singh, Hien C. Nguyen, Bharatsinai Peddi, Kriti Bhatt, Naresh Ravendranathan, Jefferson C. Frisbee, Krishna K. Singh

**Affiliations:** 1Department of Medical Biophysics, Schulich School of Medicine and Dentistry, University of Western Ontario, 1151 Richmond St. N., London, ON N6A 3K7, Canada; sbu4@uwo.ca (S.B.); asing945@uwo.ca (A.S.); hnguy29@uwo.ca (H.C.N.); bpeddi@uwo.ca (B.P.); kbhatt23@uwo.ca (K.B.); nravendr@uwo.ca (N.R.); jfrisbee@uwo.ca (J.C.F.); 2Anatomy and Cell Biology, Schulich School of Medicine and Dentistry, University of Western Ontario, London, ON N6A 3K7, Canada

**Keywords:** PDIA-4, autophagy, apoptosis, endothelial-to-mesenchymal transition, endothelial function, cardiovascular diseases

## Abstract

Endothelial autophagy plays an important role in the regulation of endothelial function. The inhibition of endothelial autophagy is associated with the reduced expression of *protein disulfide isomerase 4* (*PDIA-4*); however, its role in endothelial cells is not known. Here, we report that endothelial cell-specific loss of PDIA-4 leads to impaired autophagic flux accompanied by loss of endothelial function and apoptosis. Endothelial cell-specific loss of PDIA-4 also induced marked changes in endothelial cell architecture, accompanied by the loss of endothelial markers and the gain of mesenchymal markers consistent with endothelial-to-mesenchymal transition (EndMT). The loss of PDIA-4 activated TGFβ-signaling, and inhibition of TGFβ-signaling suppressed EndMT in *PDIA-4*-silenced endothelial cells in vitro. Our findings help elucidate the role of PDIA-4 in endothelial autophagy and endothelial function and provide a potential target to modulate endothelial function and/or limit autophagy and EndMT in (patho-)physiological conditions.

## 1. Introduction

Autophagy or “self-eating” is a highly conserved cellular process present in all eukaryotic cells. It clears damaged proteins or organelles and recycles cellular materials for reuse. Endothelial cells line the innermost layer of blood vessels and have critical roles in maintaining vascular homeostasis [1]. Imbalanced autophagy can lead to endothelial dysfunction and cardiovascular diseases [2,3,4,5,6]. Autophagy is a tightly regulated process in endothelial cells, and its genetic and epigenetic regulation is well established [3]. We have previously shown that the genetic inhibition of autophagy through the loss of *autophagy-related gene 7* (*ATG7*) leads to impaired autophagy and induces TGF-β-mediated endothelial-to-mesenchymal transition (EndMT) in vitro and in vivo [7]. During EndMT, endothelial cells acquire a migratory phenotype and begin to express different mesenchymal markers, including fibroblast-specific protein-1 (FSP-1), N-Cadherin, and α smooth muscle actin (α SMA), while downregulating endothelial cell markers including platelet endothelial cell adhesion molecule (CD31) and tek tyrosine kinase, endothelial (Tie-2) [8,9]. Finally, the cells lose endothelial phenotype and gain a mesenchymal phenotype with increased motility and contractile activities [10,11].

Previously, the reduced expression of *protein disulfide isomerase 4* (*PDIA-4*) was reported in endothelial cells following pharmacologic inhibition of autophagy [12]. Protein disulfide isomerases (PDIs) function as oxidoreductases and as chaperons that assist with the formation of disulfide bonds to help proteins fold into their final conformation [13]. PDIA-4 belongs to the PDI family and has been shown to participate in platelet activation, endoplasmic reticulum stress response (ERS response), DNA repair machinery and tumor progression [14]. ERS response is believed to be a “self-protection” mechanism in response to stress, for example, the accumulation of misfolded proteins [14]. This response can stimulate the formation of autophagosomes and trigger autophagy to clear damaged proteins [15]. It is reported that inhibiting PDIA-4 in brain cancer leads to autophagy-mediated ferroptosis [16]; however, the role of PDIA-4 in endothelial autophagy, apoptosis and function is not studied.

In the present study, we hypothesized that PDIA-4 is an essential regulator of endothelial function and survival. We demonstrate that the loss of PDIA-4 inhibits autophagy and endothelial function, and promotes apoptosis and TGFβ-mediated EndMT in endothelial cells. These data suggest that PDIA-4 may be an important and novel target to modulate endothelial function, autophagy and EndMT in (patho-)physiological conditions.

## 2. Results

### 2.1. Loss of PDIA-4 Inhibits Autophagy in Endothelial Cells

To investigate the role of *PDIA-4* in the endothelium, endothelial cells were transfected with either si*PDIA-4* or scrambled control for 24, 48 or 72 h, and cells were collected for RNA and protein extraction to perform qPCR and Western blot, respectively. Our qPCR and Western blot data on PDIA-4 confirmed successful silencing at both the transcript (Figure 1A) and the protein (Figure 1B) level, respectively. We next tested whether PDIA-4 affects autophagic activity in endothelial cells. Autophagic flux is commonly evaluated by measuring the ratio between LC3-II and LC3-I protein levels, where a lower LC3-II/LC3-I ratio indicates autophagic impairment [7]. Our Western blot data showed a gradual decrease in LC3-II levels and an increase in LC3-I levels, with an overall reduced LC3-II/LC3-I ratio indicating impaired autophagy in *PDIA4*-silenced endothelial cells in comparison to control endothelial cells (Figure 1C,D).

### 2.2. Loss of PDIA-4 Impairs Endothelial Functions

Angiogenesis, migration, and proliferation are essential functions in endothelial cells that also play an integral role in the maintenance of optimal vascular tone [17]. The in vitro angiogenesis assay is a surrogate for studying angiogenesis in cultured endothelial cells [18]. The loss of PDIA-4 resulted in a significant reduction in mesh area (Figure 2A,B), number of nodes per field (Figure 2C) and tube length per field (Figure 2D), overall indicating reduced angiogenic potential in comparison to control endothelial cells. Migratory capacity is another essential function of endothelial cells and our migration assay showed significantly reduced migration of *PDIA-4*-silenced endothelial cells in comparison to control endothelial cells (Figure 2E,F). A proliferation assay was performed to assess the proliferative ability of endothelial cells. The loss of PDIA-4 resulted in a significant reduction in cell proliferation compared to control endothelial cells (Figure 2G). In summary, the loss of PDIA-4 led to reduced endothelial function in comparison to control endothelial cells.

### 2.3. Effect of Loss of PDIA-4 on Regulators of Endothelial Functions

Nitric oxide (NO) produced by the activated endothelial NO synthase (eNOS, ser1177) is a fundamental determinant of endothelial function and Akt mediates the activation of eNOS. Endothelial cells constitutively express eNOS [19]. Accordingly, we next measured eNOS and AKT expression and activation. To our surprise, the loss of PDIA-4 did not show any significant difference in eNOS or AKT expression and activation between *PDIA-4*-silenced and control endothelial cells (Appendix A). To understand the mechanism behind the observed reduction in cellular proliferation in *PDIA-4*-silenced endothelial cells, we measured the expression level of the cell cycle inhibitor p21 in endothelial cells [20]. However, again, to our surprise, p21 was significantly down-regulated in *PDIA-4*-silenced endothelial cells in comparison to control endothelial cells (Figure 2H,I). Overall, these data indicate that the loss of *PDIA-4*-associated impaired endothelial function was independent of eNOS expression and activation.

### 2.4. Loss of PDIA-4 Induced p53-Mediated Apoptosis in Endothelial Cells

Apoptosis is an alternate cell death system to autophagy [21]. Instead of recycling selective organelles and cellular materials for future use, apoptosis destroys unwanted cells and results in cell death or bursting of the cells. It is known that autophagy and apoptosis can be triggered by the same stress signals and that there is crosstalk between the critical proteins involved in these two processes [22]. We observed impaired autophagy in *PDIA-4*-deficient endothelial cells, which was further associated with reduced cell proliferation. We next measured apoptosis in *PDIA-4*-silenced endothelial cells by measuring cleaved caspase-3 levels. Our Western blot data showed an increased level of cleaved caspase-3 protein, indicating increased apoptosis in *PDIA-4*-silenced endothelial cells in comparison to the controls (Figure 3A,B). P53 protein is an essential regulator of apoptosis [23], and to test whether the increased apoptosis in *PDIA-4*-silenced endothelial cells is mediated by p53, we measured p53 levels in *PDIA-4*-silenced and control endothelial cells. We observed a significant up-regulation of p53 expression in *PDIA-4*-silenced in comparison to control endothelial cells, indicating p53-mediated apoptosis in *PDIA-4*-silenced endothelial cells (Figure 3C,D). Furthermore, p53 is a transcription factor, and it induces apoptosis by regulating downstream targets like pro-apoptotic Bax and by shifting the ratio of the level of Bax to that of the pro-survival molecule Bcl-2 [24,25]. Accordingly, we measured the expression level of Bax and Bcl-2 in *PDIA-4*-silenced and control endothelial cells. In line, we observed an increase in Bax expression and reduced Bcl-2 expression in *PDIA-4*-silenced endothelial cells (Figure 3E–G), shifting the overall balance from pro-survival to pro-apoptosis in *PDIA-4*-silenced endothelial cells.

### 2.5. Loss of PDIA-4 Induced EndMT in Endothelial Cells

We have previously reported that the loss of ATG7 in endothelial cells leads to impaired autophagy and EndMT in vitro and in vivo [7]. We next reported a reduced expression of PDIA-4 following pharmacologic inhibition of autophagy in endothelial cells [12]. In this study, we have shown that the loss of PDIA-4 is associated with autophagy inhibition, which is in line with the loss of ATG7-associated autophagic inhibition. Next, to clarify whether the loss of PDIA-4 is also associated with autophagic inhibition-associated EndMT, we evaluated EndMT in *PDIA-4*-silenced and control endothelial cells. Surprisingly, *PDIA-4*-deficient endothelial cells imaged under a phase contrast microscope displayed a conspicuous transition from the distinctive cobblestone-like appearance of endothelial cells to an enlarged spindle-shaped pattern that is consistent with fibroblast-like morphology of mesenchymal cells, indicating EndMT in *PDIA-4*-silenced endothelial cells (Figure 4A) [7]. To confirm EndMT at the molecular level, the expression levels of endothelial cell markers CD31 and Tie-2, and mesenchymal markers αSMA, N-Cadherin and FSP-1 were measured in *siPDIA-4*-transfected vs. scrambled control-transfected endothelial cells. CD31 maintains endothelial cell junction integrity [26]; Tie-2 belongs to the receptor tyrosine kinase family [27]; αSMA is involved in cytoskeletal reorganization during EndMT [28]; FSP-1 promotes cell differentiation and motility [29] and N-Cadherin serves as anchor points during cell-cell contact [30]. Expectedly, the loss of endothelial and gain of mesenchymal phenotype was mirrored; reduced expression of CD31 and Tie-2 (Figure 4B–F) and increased expression of mesenchymal markers αSMA, N-Cadherin, and FSP-1 (Figure 4G–M) was observed in *PDIA-4*-silenced cells in comparison to control endothelial cells. These imaging and molecular data confirmed that loss of PDIA-4 in endothelial cells induced EndMT.

### 2.6. Loss of PDIA-4 Induced EndMT Is Mediated by the TGFβ Pathway

TGFβ signaling activation is the main stimulator of EndMT in endothelial cells [31]. Upon binding of TGFβ1 to TGFβ receptors, TGFBR1, TGFBR2, and SMAD2 are phosphorylated and interact with SMAD3 to translocate to the nucleus and activate transcription of genes involved in EndMT [32]. Accordingly, to evaluate whether TGFβ mediates loss of *PDIA-4*-induced EndMT, the expression level of the ligand TGFβ1 was measured, which demonstrated a significant upregulation in *PDIA-4*-silenced endothelial cells in comparison to control endothelial cells (Figure 5A). We then measured the expression level of TGFβ1 receptors TGFBR1 and TGFBR2 and observed a significantly increased expression of these receptors in *PDIA-4*-silenced endothelial cells in comparison to the control endothelial cells (Figure 5B,C). Next, to confirm whether the increased expression of TGFβ1 and its receptors leads to the increased activation of TGFβ signaling, we measured the activated level of SMADs in *PDIA-4*-silenced endothelial cells. We observed an increased expression level of SMAD3 and SMAD4 and an increased activation level of SMAD3 in the TGFβ signaling pathway (Figure 5D–F). We then measured transcription factor SLUG, which is a downstream effector in the TGFβ signaling pathway and observed a significant upregulation in *PDIA-4*-silenced endothelial cells compared to the controls (Figure 5G). We also measured TGFβ-responsive pro-fibrotic genes, such as CTGF and Collagen I, and as expected, we observed a significant upregulation of both genes in the *PDIA-4*-silenced endothelial cells in comparison to control endothelial cells (Figure 5H,I). Increased Collagen I expression in *PDIA-4*-silenced endothelial cells was also confirmed at the protein level (Figure 5J,K). To further confirm if loss of *PDIA-4*-induced EndMT is mediated by TGFβ signaling, we inhibited TGFβ signaling and measured the expression level of EndMT markers in *PDIA-4*-silenced and control endothelial cells. TGFβ signaling inhibitor SB431542 is a potent and competitive inhibitor that inactivates the SMAD2/3 complex and blocks TGFβ signaling [33]. Inhibitor SB431542 treatment to the *PDIA-4*-silenced and control endothelial cells blocked EndMT in *PDIA-4*-silenced endothelial cells, which is evident by restored expression of endothelial marker CD31 (Figure 5L) and reduced expression of mesenchymal marker N-Cadherin (Figure 5M,N) in *PDIA-4*-silenced endothelial cells. Overall, these data confirm that TGFβ signaling mediates EndMT in *PDIA-4*-silenced endothelial cells.

## 3. Discussion

The main observation made in this study is that loss of endothelial PDIA-4 impairs autophagic flux and endothelial function and induces TGFβ-mediated EndMT. These data suggest that PDIA-4 may be a potentially novel and previously unrecognized target to modulate autophagy, endothelial function and EndMT in endothelial cells to limit impaired autophagy, endothelial dysfunction and/or EndMT-associated diseases.

We previously reported that the loss of ATG7 in endothelial cells is associated with the inhibition of autophagy and the induction of EndMT in endothelial cells [7]. We then reported a reduced expression of PDIA-4 in endothelial cells following pharmacologic inhibition of autophagy [12]. We continued this line of investigation, and now we report that loss of PDIA-4 inhibits autophagy and induces EndMT. The endoplasmic reticulum (ER) is known to be related to autophagy as it contributes to the formation of autophagosomes [34]. One similarity between the ER and autophagy is that they are both involved in clearing waste products in the cells. Specifically for the ER, the accumulation of improperly folded protein serves as a stress signal and triggers ER stress response (ERS response) and autophagy to remove damaged proteins [35]. PDIA-4 functions as a protein chaperon and is upregulated during the ERS response [14]. There is no information linking PDIA-4 to endothelial autophagy, although Quan et al. found that autophagy-deficient β-cells were susceptible to ERS response and may contribute to diabetes [36]. Offsprings obtained from obese mice interbred with mice autophagy-deficient in β-cells develop severe diabetes due to increased apoptosis in β cells [36]. Overall, this is the first report directly linking PDIA-4 with autophagy.

Similarly, we, for the first time, report that loss of PDIA-4 is associated with impaired endothelial function, with reduced angiogenic potential, cell migration, and cell proliferation. There is no known direct link between autophagy and endothelial function; however, matrix proteoglycans like Decorin and Perlecan provide an indirect link [37] as both of these proteoglycans can regulate angiogenesis and autophagy [38,39]. Xing et al. found that loss of PDIA-4 in cervical cancer cells significantly impaired cell proliferation and cell migration [40]. Additionally, it was found that the knockdown of PDIA-4 resulted in significantly lower cell proliferation in glioma cells [41]. The inhibition of PDIA-1, an isoform of PDIA-4, in breast cancer cells and endothelial cells resulted in a significantly slower migration rate compared to the control [42]. Endothelial dysfunction is closely linked to atherosclerosis, mainly by making cells more susceptible to ERS response [43]. Civelek et al. found a significant upregulation of genes involved in ERS response in athero-susceptible regions, linking endothelial dysfunction, ERS response and atherosclerosis [43]. Protein p21 is a cell cycle inhibitor [20], and we expected an increased p21 expression in *PDIA-4*-silenced endothelial cells. It is important to note that p21 also plays context-dependent roles, such as in apoptosis [44]. Here, we observed a reduced protein level of p21 in *PDIA-4*-silenced endothelial cells compared to control cells, indicating p21-independent inhibition of the cell cycle in *PDIA-4*-silenced endothelial cells. It is possible that reduced cell proliferation due to the loss of *PDIA-4* occurred through a p21-independent pathway.

Apoptosis often crosstalk with autophagy, and given the reduced angiogenic potential, migration, and proliferation, which were not associated with regulators of endothelial function such as eNOS and AKT expression/activation, we next measured apoptosis. Interestingly, we did observe an increased apoptosis in *PDIA-4*-silenced endothelial cells. Pro-apoptotic p53 is an essential regulator of apoptosis [23]. We observed an increased expression of p53 following the loss of *PDIA-4* in endothelial cells. Bcl-2 is a pro-survival and Bax is a pro-apoptotic molecule, and both of them are downstream effectors of p53 in a context-dependent manner [25]. We observed increased Bax and reduced Bcl-2 levels, indicating increased p53-mediated apoptosis in *PDIA-4*-deficient endothelial cells. Overall, these results show that reduced endothelial function in *PDIA-4*-silenced endothelial cells is mainly attributed to increased apoptosis, indicating an important role played by PDIA-4 in maintaining endothelial cell function and survival.

We have previously reported that the loss of ATG7 results in impaired autophagy and TGFβ-mediated EndMT in endothelial cells [7]. We also reported a reduced expression of PDIA-4 following autophagy inhibition in endothelial cells [12]. We then investigated whether loss of *PDIA-4*, which inhibits autophagy, would also induce EndMT. As expected, silencing *PDIA-4* indeed induced EndMT, which was associated with the transition of endothelial cells from their characteristic cobblestone outlook to an elongated fibroblastic morphology. These findings were further confirmed by measuring the expression of endothelial and mesenchymal markers. Furthermore, TGFβ stimulates EndMT in autophagy-deficient ECs [7,45]. Accordingly, we observed an increased expression of ligand TGFβ1; its receptors TGFBR1, TGFBR2; effectors SMAD3, SMAD4; downstream transcription factor SLUG and TGFβ-responsive gene CTGF and collagen I in *PDIA-4*-silenced compared to control endothelial cells, suggesting that loss of *PDIA-4* induced EndMT through a TGFβ-dependent pathway (Figure 6). This was further confirmed by utilizing the pharmacologic inhibition of the TGFβ pathway, which restored endothelial marker CD31 and mesenchymal marker N-Cadherin expression in *PDIA-4*-silenced ECs. These results are supported by the findings from Singh et al. that genetic (via loss of *ATG7*) inhibition of autophagy leads to TGFβ-mediated EndMT in endothelial cells. As mentioned previously, EndMT is associated with the pathology of many types of organ fibrosis, including pulmonary, cardiac, kidney and cancer fibrosis, mainly resulting from increased expression of extracellular proteins such as collagen [46,47,48]. The results from this study warrant further investigation into the therapeutic potential of PDIA-4 for treating impaired autophagy-, impaired endothelial dysfunction-, and/or EndMT-associated diseases, such as atherosclerosis and organ fibrosis.

## 4. Materials and Methods

Cell Culture—Human umbilical vein endothelial cells [HUVECs [passage 5–7, (Lonza, Rockville, MD, USA)]], a common in vitro standard model to study endothelial cells [8,49,50] were cultured in endothelial growth medium (EGM-2, Lonza, Rockville, MD, USA) supplemented with growth factors, serum and antibiotics at 37 °C in humidified 5% CO_2_ incubator. HUVECs were transfected with 5 nm *PDIA-4* silencing RNA (si*PDIA-4*) (Santa Cruz, CA, USA: Catalog # sc-44571) using a standard reverse transfection protocol (Lipofectamine ^®^ 3000, Thermo Scientific, Waltham, CA, USA). RNAs or proteins were collected 24, 48 or 72 h post-transfection. TGFβ inhibitor experiments were conducted using the drug SB431542 (10 μM). Cells were reverse transfected as previously described [49], incubated in complete media for 72 h, and then treated with 10 μM of SB431542 for another 48 h.

### 4.1. cDNA Synthesis and Real-Time Quantitative PCR

To measure the expression level of different genes, quantitative real-time PCR (RT-qPCR) was used. Total RNA was extracted from HUVECs in Trizol reagent (Invitrogen, Carlsbad, CA, USA). RNA concentration was measured using the NanoDrop (Thermo Scientific, Madison, WI, USA). Complementary DNA (cDNA) was synthesized using the Quantitect Reverse Transcription Kit (Qiagen, Hilden, Germany), followed by a polymerase chain reaction using a Thermal Cycler Real-Time PCR machine (Eppendorf, Eppendorf, Germany). RT-qPCR was conducted with the mixture of samples’ RNA, SYBR Select Master Mix (Applied Biosystems, Vilnius, Lithuania) and forward and reverse primers of *PDIA-4*, *Cyclin-dependent Kinase Inhibitor 1a (P21)* [49], *Transforming Growth Factor-beta Receptor*, *Type 1* (*TGFBR1)*, *TGFBR2*, *SMAD3*, *SMAD4*, *SLUG*, *Transforming Growth Factor β (TGF β)*, *Fibroblast Specific Protein 1 (FSP-1)*, *Tek Tyrosine Kinase, Endothelial (Tie-2), N-Cadherin*, *CD31*, *Connective Tissue Growth Factor (CTGF)*, *Collagen I* and *GAPDH* [9] (Table 1). RT-qPCR was run using the QuantStudio^®^3 Real-Time PCR Instrument (Applied Biosystems, Singapore). The comparative *Delta Delta CT* method was employed for data analysis [51].

### 4.2. Western Blotting

Cell lysates were first collected and extracted in RIPA buffer (Millipore, Burlington, MA, USA) 24, 48, and 72 h post-transfection with either siPDIA-4 or scrambled controls. Equal amounts of protein were loaded onto SDS-polyacrylamide gels and transferred to the PVDF membranes (Thermo Fisher, Madison, WI, USA). Membranes were blocked for 1 h in 1X TBS with 3% BSA and incubated overnight at 4 °C with primary antibodies. The primary antibodies used in this study include p21 (Cell Signaling #2947S), Cleaved Caspase-3 (Cell Signaling #9664S), LC3 I/II (Cell Signaling #4600IAP), GAPDH (Cell Signaling #5174S), p-SMAD3 (ABclonal #51451), SMAD3 (ABclonal #28379), Βeta Actin (Cell Signaling #4970S), Βeta Tubulin (Proteintech Cat#66240-1-Ig), Bax (Cell Signaling #2772S), Bcl-2 (Cell Signaling #3498S), CD31 (ABclonal Cat#A4900), α smooth muscle actin (ABclonal Cat#A17910), N-Cadherin (ABclonal Cat#A19083), PDIA-4 (Cell Signaling #5033S), and P53 (Santa Cruz #sc-126). Proteins were then incubated with goat anti-rabbit secondary antibody (Enzo ADI-SAB-300-J) for 2 h at room temperature. Bands were visualized with the ECL substrates using a chemiluminescence channel and 700 channels in the LiCor Fc Odyssey system. The bands were quantified using LiCor Fc Odyssey inbuilt software version 5.2.

### 4.3. Proliferation Assay

To analyze the proliferation of HUVECs after transfection with siPDIA-4 and the scrambled controls, a colorimetric proliferation assay with WST-1 reagent was performed according to instructions from the manufacturer (Roche Applied Science, Rotkreuz, Switzerland, Cat. No. 11644807001). Approximately 1.5 × 10^5^ cells/mL concentration *per* well were seeded in the 6-well assay plate and transfected with siPDIA-4 and scrambled control as described before. Cells were incubated with warm EGM-2 at 37 °C with 5% CO_2_ for 72 h and collected with trypsin; 100 μL cells were reseeded in triplicates for each biological replicate into 96 well plates, and 10 μL of WTS-1 reagent was added to each well. Cells were incubated with warm EGM-2 at 37 °C with 5% CO_2_ for 4 h and absorbance was measured at 440 nm. WST-1 is a red tetrazolium salt and is cleaved by metabolically active cells to formazan, which is dark red; by quantifying the amount of formed formazan, the amount of proliferating cells can be measured [52].

### 4.4. Migration Assay

Briefly, 1.2–1.5 × 10^5^ cells/mL concentration *per* well were seeded in the 6-well assay plate and transfected with *siPDIA-4* or scrambled control as described previously. Transfected cells were incubated with warm EGM-2 media at 37 °C with 5% CO_2_ for 24 h or until 90% confluency was reached. A straight scratch was made in the middle of the cell monolayer using a p200 pipette tip, followed by washing with 1X PBS and incubation with low-serum (1%) media [53]. Cells were imaged immediately at T_0_ using phase contrast microscopy. Cells were then put back into the incubator and imaged every 4 h up to 20 h. The percent of open wound area at each time point was calculated using the ImageJ version 1.54 [54].

### 4.5. Angiogenesis Assay

The in vitro Angiogenesis Assay Kit (Millipore #ECM625, Darmstadt, Germany) was used as instructed by the manufacturer [49]. Briefly, transfected cells were cultured on the 96-well plate coated with ECMatrix provided by the kit and subsequently evaluated for tube formation abilities as instructed by the manufacturer. Cells were imaged at multiple time points. Tubes were quantified using the Angiogenesis tool on ImageJ.

### 4.6. Immunofluorescence

Immunofluorescence experiments were carried out in 4-chamber microscopy slides performed as previously described [9]. Immunofluorescence signals from CD31 (Cell Signaling #3528S) and αSMA (Cell Signaling #19245S) staining were visualized using Alexa fluor-tagged anti-mouse (Fisher Scientific #A21202) and Alexa fluor-tagged anti-rabbit (Fisher Scientific #A11034) secondary antibody, respectively. Fluorescent microscopy images were captured using the Zeiss LSM700 confocal microscope, and ZEN imaging software version 3.1 was utilized for image processing.

### 4.7. Statistical Analysis

Data are expressed as the mean ± SD. Student’s *t*-test was applied when the means of two groups were compared. An ANOVA with Tukey’s post-hoc correction was applied when the means of more than two groups were compared using GraphPad-Prism software. A *p*-value < 0.05 was considered to indicate statistical significance.

## Figures and Tables

**Figure 1 ijms-25-03913-f001:**
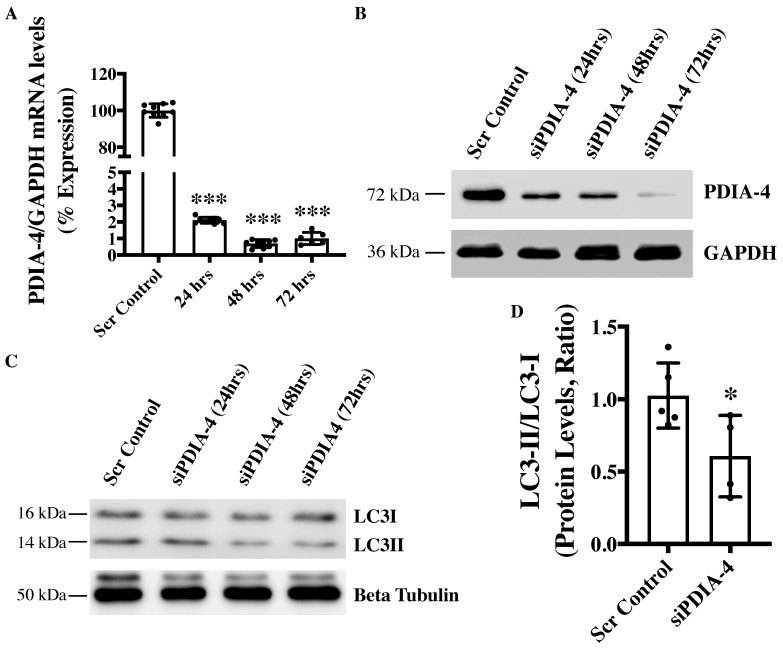
Loss of *PDIA-4* inhibits autophagy in endothelial cells. HUVECs were reverse transfected with either si*PDIA-4* or scrambled control for 24, 48 or 72 h. RNA and proteins were extracted, and (**A**) qPCR was conducted targeting *PDIA-4* and GAPDH (*n* = 3 in triplicate). (**B**) Western blot was performed for PDIA-4 and GAPDH as loading control. (**C**) Western blot was performed targeting LC3 and beta tubulin (*n* = 4–5). (**D**) Quantification of the Western blot data 72 h post-transfection. Data were analyzed using the Student’s *t*-test, and data are expressed as mean ± SD. * and *** represent *p* < 0.05 and <0.001, respectively, vs. respective Scr Cont.

**Figure 2 ijms-25-03913-f002:**
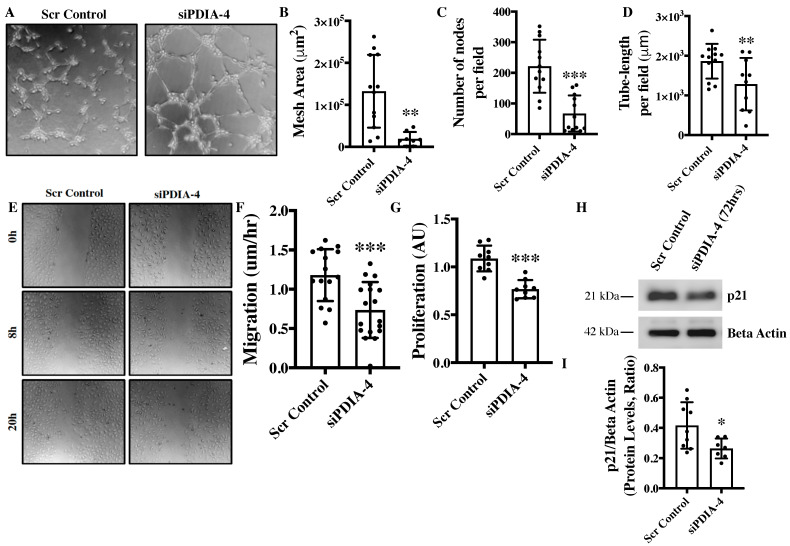
Loss of PDIA-4 impairs endothelial functions. HUVECs were transfected with either si*PDIA-4* or scrambled control for 72 h. (**A**–**D**) Cells were extracted and seeded on 96-well plates on Matrigel according to the manufacturer’s guidelines. Cells were imaged after 4 h incubation at 37 °C in a humidified 5% CO_2_ incubator. Tubes were quantified using the ImageJ angiogenesis tool (*n* = 3 in triplicates). (**E**,**F**) A scratch was made using a p200 pipette tip in the middle of the cell monolayer, and cells were incubated in low serum media (MCDB131 + 1% FBS). Cells were imaged immediately at T_0_ using phase contrast microscopy, then put back into the incubator and imaged every 4 h up to 20 h. The percentage of open wound area at each time point was calculated using the ImageJ wound healing tool. Migration (the velocity of the cells to close the entire scratch area) was calculated (*n* = 15–17). (**G**) HUVECs were transfected with siPDIA-4 or scrambled control into 96-well plates for 72 h (*n* = 9), and 10 μL of WST-1 reagent was added to each well. Cells were incubated with warm EGM2 at 37 °C and 5% CO_2_ for 4 h and absorbance was measured at 440 nm. HUVECs were reverse transfected with si*PDIA-4* or scrambled control and incubated in complete EGM-2 medium for 72 h. Protein was extracted and (**H**) Western blot was performed targeting p21 and beta actin (loading control) (*n* = 7–9). (**I**) The bands were quantified using LiCor Fc Odyssey inbuilt software. Data were analyzed using Student’s *t*-test and are expressed as mean ± SD. *, ** and *** represent *p* < 0.05, *p* < 0.01 and <0.001, respectively, vs. respective vehicle or Scr Cont.

**Figure 3 ijms-25-03913-f003:**
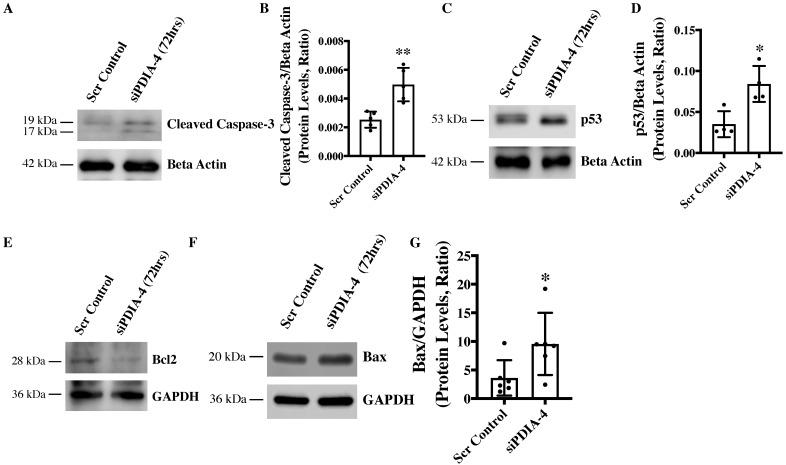
Loss of *PDIA-4*-induced p53-mediated apoptosis in endothelial cells. HUVECs were reverse transfected with si*PDIA-4* or scrambled control for 72 h. Proteins were extracted and Western blot was performed targeting (**A**) cleaved caspase-3 and beta actin (loading controls), (**C**) p53 and beta actin (loading controls), (**E**) Bcl2, (**F**) Bax and GAPDH (loading controls). (**B**,**D**,**G**) The bands were quantified using LiCor Fc Odyssey inbuilt software (*n* = 5–6). Data were analyzed using Student’s *t*-test (*n* = 3–4). Data are expressed as mean ± SD. * and ** represent *p* < 0.05 and <0.01, respectively, vs. respective vehicle or Scr Cont.

**Figure 4 ijms-25-03913-f004:**
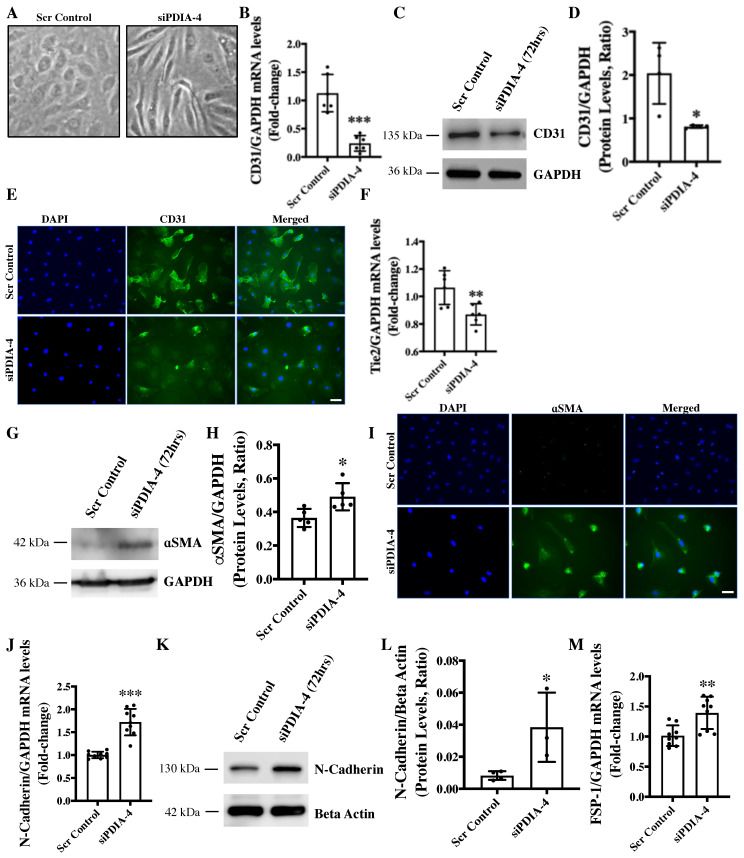
Loss of *PDIA-4*-induced EndMT in endothelial cells. (**A**) HUVECs were reverse transfected with si*PDIA-4* or scrambled control for 72 h. Images were taken under a phase contrast microscope. RNA was extracted and qPCR was performed for (**B**) CD31 (**F**) Tie-2, (**J**) N-Cadherin and (**M**) FSP-1 (*n* = 2–3 in triplicates). Proteins were extracted, and Western blot was performed targeting (**C**) CD31, (**G**) α SMA and (**K**) N-Cadherin (*n* = 3–5). (**D**,**H**,**L**) The bands were quantified using the LiCor Fc Odyssey inbuilt software. HUVECs were reverse transfected with si*PDIA-4* or scrambled control for 72 h on 4-chambers slides and immunofluorescence was performed for (**E**) CD31 (green) and (**I**) αSMA (green). Bar = 100 μm. Data were analyzed using Student’s *t*-test. Data are expressed as mean ± SD. *, ** and *** represent *p* < 0.05, <0.01 and <0.001, respectively, vs. Scr Cont.

**Figure 5 ijms-25-03913-f005:**
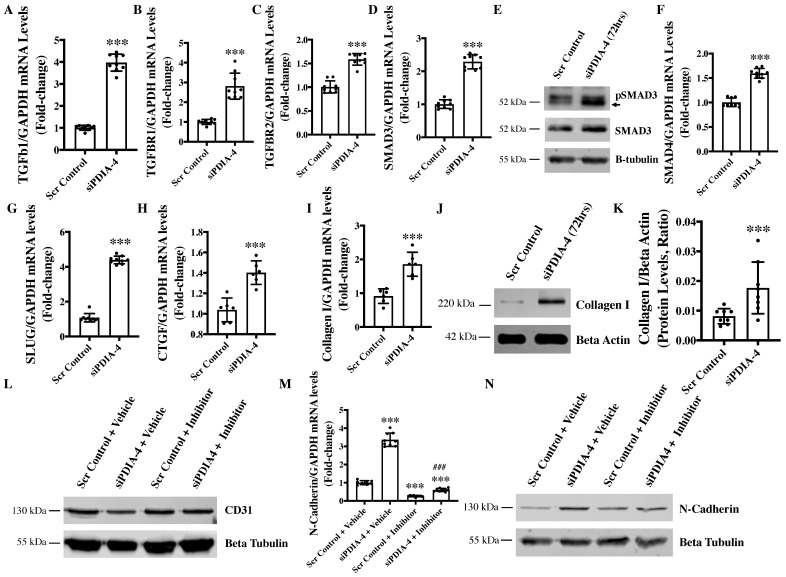
Loss of *PDIA-4* induced TGFβ-mediated EndMT. HUVECs were reverse transfected with si*PDIA-4* or scrambled control and incubated in complete EGM-2 medium for 72 h. RNA was extracted, and qPCR was performed for (**A**) TGFβ1, (**B**) TGFBR1, (**C**) TGFBR2, (**D**) SMAD3, (**F**) SMAD4, (**G**) SLUG, (**H**) CTGF, (**I**) Collagen I and GAPDH (*n* = 3 in triplicates). HUVECs were reverse transfected with si*PDIA-4* or scrambled control and incubated in complete EGM-2 medium for 72 h and proteins were extracted to perform Western blot for I SMAD3 and pSMAD3, and (**J**) collagen I. (**K**) Collagen I expression was quantified by densitometry (*n* = 7). After transfecting with si*PDIA-4* for 72 h, HUVECs were treated with either DMSO as vehicle control or TGFβ inhibitor SB431542 (10 μM). Western blot was performed targeting (**L**) CD31 (*n* = 3) and (**N**) N-Cadherin (*n* = 3). qPCR was performed targeting (**M**) N-Cadherin (*n* = 3 in triplicates). Data were analyzed using Student’s *t*-test for all figures except (**M**), where one-way ANOVA with Tukey’s post-hoc analysis was performed. Data are expressed as mean ± SD. In (**A**–**I**), *** represents *p* < 0.001 vs. Scr Cont. In (**M**), *** represents *p* < 0.001 vs. Scr Cont + vehicle, and ### represents *p* < 0.001 vs. Scr Cont + inhibitor (*n* = 3 in triplicates).

**Figure 6 ijms-25-03913-f006:**
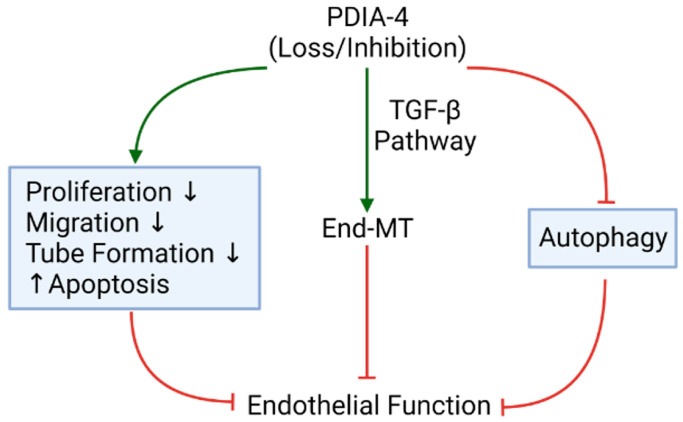
Graphical Summary. Effect of PDIA-4 loss or inhibition on endothelial function, endothelial autophagy, and endothelial survival. Downward and upward-facing black arrows indicate “inhibition” and “activation”, respectively. Red indicates “inhibition” and green arrows indicate “activation”. Created with BioRender.com accessed on 14 March 2024.

**Table 1 ijms-25-03913-t001:** List of primers used to amplify genes. Hsa: *Homo sapiens*, F: forward and R: reverse.

Primer Name	Sequences (5′-3′)
Hsa*-PDIA-4-F*	5′-CACGCTTGTGTTGACCAAAGA-3′
Hsa*-PDIA-4-R*	5′-AATTGGAGGAGAACGCTTGCT-3′
Hsa*-TGFB1-F*	5′-TACAGCACGGTATGCAAGCC-3′
Hsa*-TGFB1-R*	5′-GCAACCGATCTAGCTCACAGAG-3r′
Hsa*-SMAD3-F*	5′-TGGACGCAGGTTCTCCAAAC-3′
Hsa*-SMAD3-R*	5′-CCGGCTC GCAGTAGGAAC-3′
Hsa*-SMAD4-F*	5′-CTCATGTGATCTATGCCCGTC-3′
Hsa*-SMAD4-R*	5′-AGGTGATACAACTCGTTCGTAGT-3′
Hsa*-CD31-F*	5′-TCCTGAGGGTCAAGGTAATAGC-3′
Hsa*-CD31-R*	5′-CTCCAGACTGTACATCGTTACCC-3′
Hsa*-VE-Cadherin-F*	5′-GAGTCTTCAGCTCACGGACA-3′
Hsa*-VE-Cadherin-R*	5′-CAGCTCTGTGAGCCTCTGC-3′
Hsa*-SLUG-F*	5′-ATCTGCGCiCAAGGCGTTTTCCA-3′
Hsa*-SLUG-R*	5′-GAGCCCTCAGATTTGACCTGTC-3′
Hsa*-TGFBR1-F*	5′-TACCTGAACCCGTGTTGCTCTC-3′
Hsa*-TGFBR1-R*	5′-GTTGCTGAGGTATCGCCAGGAA-3′
Hsa*-TGFBR2-F*	5′-GTCTGTGGATGACCTGGCTAAC-3′
Hsa*-TGFBR2-R*	5′-GACATCGGTCTGCTTGAAGGAC-3′
Hsa*-FSP1-F*	5′-TCTTGGTCTGGTCTCAACGG-5′
Hsa*-FSP1-R*	5′-TGTCACCCTCTTTGCCTGAG-3′
Hsa*-N-Cadherin-F*	5′-CCTCCAGAGTTTACTGCCATGAC-3′
Hsa*-N-Cadherin-R*	5′-GTAGGATCTCCGCCACTGATTC-3′

## Data Availability

Data are available upon request to the corresponding author.

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
