# Peer review of "Protein Disulfide Isomerase 4 Is an Essential Regulator of Endothelial Function and Survival"

_ijms, 2024, doi:10.3390/ijms25073913_

Round 1

Reviewer 1 Report

Comments and Suggestions for Authors

1. The format of this manuscript should be reorganized according to the author guidelines of IJMS.

2. The panel labeling of figures should be revised.

3. In Figures 1B & C, all the time points of PDIA-4 siRNA group should be performed on the same blot.

4. In Figures 1E, the LC3I/II ratio was not significant. Please re-check these data. In addition, more evidence for the activation of autophagy should be provided.

5. In Figure 3, the results of caspase-3 cannot support the conclusion of this study. The change in the levels of cleaved caspase-3 was not obvious. In addition, the caspase-3 and cleaved caspase-3 should be shown on the same blot.   

6. More evidence for the apoptosis and activation of related signal pathways should be provided.

7. The image quality of Figure 2A and 4A was low, please use high-quality of images to replace the original ones.

8. In Figure 4H, the trend in the changes of α-SMA (quantitative data) was not consistent with that of the images (Figure 4I).

9. More functional evidence for End-MT is required for this study.

Comments on the Quality of English Language

 Minor editing of English language required

Author Response

Reviewer 1

  1. The format of this manuscript should be reorganized according to the author guidelines of IJMS.

Response: The manuscript has been formatted as suggested.

  1. The panel labeling of figures should be revised. 

Response: The panel labeling of figures is revised as suggested.

  1. In Figures 1B & C, all the time points of PDIA-4 siRNA group should be performed on the same blot.

Response: We have now provided a blot for PDIA4 with all time-points.

  1. In Figures 1E, the LC3I/II ratio was not significant. Please re-check these data. In addition, more evidence for the activation of autophagy should be provided.

Response: We looked into the analysis and observed a significant (p= 0.0419) difference. We are providing here the data used to generate that graph.

Scr Control    siPDIA-4

0.917468         0.414556

0.874609         0.318704

0.82223823     0.88831909

1.15126657     0.80461752

1.35901195    

Additionally, we also tried to include data on p62, which is another marker for autophagy; however, we did not get a quality data.

  1. In Figure 3, the results of caspase-3 cannot support the conclusion of this study. The change in the levels of cleaved caspase-3 was not obvious. In addition, the caspase-3 and cleaved caspase-3 should be shown on the same blot.   

Response: We now provide a blot showing cleaved caspase 3 (figure 3A).

  1. More evidence for the apoptosis and activation of related signal pathways should be provided.

Response: Our data show that loss of PDIA-4 induces p53-mediated apoptosis, and accordingly, data on p53 and p53-regulated bax and bcl-2 has been provided.

  1. The image quality of Figure 2A and 4A was low, please use high-quality of images to replace the original ones.

Response: The images have been replaced with better quality images.

  1. In Figure 4H, the trend in the changes of α-SMA (quantitative data) was not consistent with that of the images (Figure 4I).

Response: Figure 4H is the quantification of the western blot (figure 4G). The representative western blot is provided in figure 4G. Normally, we see a very minimal (or no expression) of α-SMA in endothelial cells, and accordingly we see almost absence of α-SMA in control endothelial cells via IF; however, given the amount of protein loaded and sensitivity, we observe a minimal expression in western blot.

  1. More functional evidence for End-MT is required for this study.

Response: As suggested, we now include qPCR data on collagen-1 (figure 5I), which plays a role in the functional consequence of EndMT.

Reviewer 2 Report

Comments and Suggestions for Authors

The paper by Bu S. and coworkers reports the role of protein disulfide isomerase 4 (PDIA-4) in endothelial cells on the control of autophagy and angiogenic functional responses which are crucial fo endothelial to mesenchymal-transition (End-MT) and endothelial function.

Indeed silencing/inhibition of PDIA-4 impairs autophagy and endothelial angiogenic responses while promoting End-MT.

The paper is interesting and the experimental approach is adequately designed and conducted in cultured cells.

I suggest some checks and  improvements:

1) figures should be organized in a more consisted manner. If quantification of blots or other signals is reported after the blots or pictures, please arrange them in a uniform manner. Probably the organization of figure in horizontal setting may improve the general outlook of the paper.

2) Data on apoptosis and cleaved caspase 3 in figure 2 A and B are questionable.  The difference  of lane intensity for caspase 3 in A seems not to reflect the quantification of panel B. I suggest to repeat the cleaved  caspase 3 blots considering total caspase 3 in addition to beta actin.

3) please check the consistency of hrs/hr throughout the manuscript  and space presence between the number and the unit measure.

4) some editing check is necessary as int first page organization, at  line 230 and 241, in figure legends.

5) A curiosity: is there any relation reported in the literature between autophagy and angiogenesis? if yes, please report and discuss the finding.

Comments on the Quality of English Language

The quality of English language is enough good. Some technical reporting (unit measures, spaces, etc) should be checked for uniformity

Author Response

Reviewer 2

The paper by Bu S. and coworkers reports the role of protein disulfide isomerase 4 (PDIA-4) in endothelial cells on the control of autophagy and angiogenic functional responses which are crucial for endothelial to mesenchymal-transition (End-MT) and endothelial function.

Indeed silencing/inhibition of PDIA-4 impairs autophagy and endothelial angiogenic responses while promoting End-MT.

The paper is interesting, and the experimental approach is adequately designed and conducted in cultured cells.

I suggest some checks and improvements:

1) figures should be organized in a more consisted manner. If quantification of blots or other signals is reported after the blots or pictures, please arrange them in a uniform manner. Probably the organization of figure in horizontal setting may improve the general outlook of the paper.

Response: Thanks for your positive comments. The figures are re-organized as suggested.

2) Data on apoptosis and cleaved caspase 3 in figure 2 A and B are questionable.  The difference of lane intensity for caspase 3 in A seems not to reflect the quantification of panel B. I suggest to repeat the cleaved caspase 3 blots considering total caspase 3 in addition to beta actin.

Response: As suggested, we now provide a better representative blot for cleaved caspase 3.

3) please check the consistency of hrs/hr throughout the manuscript and space presence between the number and the unit measure.

Response: It has been corrected.

4) some editing check is necessary as int first page organization, at line 230 and 241, in figure legends.

Response: First page is revised.

5) A curiosity: is there any relation reported in the literature between autophagy and angiogenesis? if yes, please report and discuss the finding.

Response: There is no known direct link between autophagy and endothelial function; however, matrix proteoglycans like Decorin and Perlecan provide an indirect link (PMID: 31428311) as both of these proteoglycans regulate angiogenesis and autophagy (PMID: 23798385, PMID: 24737315). This information has been included in the revised manuscript.

Round 2

Reviewer 1 Report

Comments and Suggestions for Authors

1. The quality of Figures 2A, 2E, and 4A was too low for publication of IJMS. Authors should provide better-quality images to replace the original ones.

2. Experimental evidence for the activation of autophagy flux and EndMT is required. In addition, functional assays and western blot analysis are highly recommended for these comments.

Author Response

The quality of Figures 2A, 2E, and 4A was too low for publication of IJMS. Authors should provide better-quality images to replace the original ones.

Response: We have already provided the best quality image available for 2A, 2E and 4A in the first revision and we are afraid that we will not be able to provide new figures in the given time-frame. 

2. Experimental evidence for the activation of autophagy flux and EndMT is required. In addition, functional assays and western blot analysis are highly recommended for these comments.

Response: As I explained in response to the first revision  that we were not able to get quantifiable data for another autophagy marker P62, and accordingly did not include that data in the manuscript. LC3-II/LC3-I ratio is the most utilized marker of autophagy and we consistently observed reduced (LC3-II/LC3-I ratio) autophagy following PDIA-4 loss in endothelial cells.  We completely agree that additional western blot and functional assays will further strengthen our conclusion, however, we have continued this line of investigation and currently  testing the effect of loss-of PDIA4-induced EndMT in fibrosis model, which will be the basis of our next publication.

Thanks agaiin!
Krishna